# Evaluation of a screening algorithm using the Strengths and Difficulties Questionnaire to identify children with mental health problems: A five-year register-based follow-up on school performance and healthcare use

**Rasmus Trap Wolf** [1,2]*, **Pia Jeppesen**[2,3], **Dorte Gyrd-Hansen**[1], **The CCC2000 Study Group**[4¶], **Anne Sophie Oxholm**[1]

1 University of Southern Denmark, Department of Public Health, Danish Centre for Health Economics, Odense, Denmark, 2 Child and Adolescent Mental Health Centre, Mental Health Services Capital Region of Denmark, Copenhagen, Denmark, 3 Department of Clinical Medicine, Faculty of Health and Medical Sciences, University of Copenhagen, Copenhagen, Denmark, 4 Centre for Clinical Research and Prevention, Capital Region of Denmark, Copenhagen, Denmark

¶ Membership of the CCC2000 Study Group is provided in the Acknowledgments.
* rasmus.trap.wolf@regionh.dk

**Data Availability Statement:** The data utilized in the current study are defined as sensitive personal

## Abstract

### Background

Treatment of mental health problems (MHP) is often delayed or absent due to the lack of systematic detection and early intervention. This study evaluates the potential of a new screening algorithm to identify children with MHP.

### Methods

The study population comprises 2,015 children from the Copenhagen Child Cohort 2000 whose mental health was assessed at age 11–12 years and who had no prior use of special-ised mental health services. A new algorithm based on the Strengths and Difficulties Questionnaire (SDQ) is utilised to identify MHP by combining parent-reported scores of emotional and behavioural problems and functional impairments. The screening is done on historical data, implying that neither parents, teachers nor health care professionals received any feedback on the screening status. The screening status and results of an IQ-test were linked to individual-level data from national registries. These national registers include records of each child's school performance at the end of compulsory schooling, their health care utilisation, as well as their parents' socio-economic status and health care utilisation.

### Results

10% of the children screen positive for MHP. The children with MHP achieve a significantly lower Grade Point Average on their exams, independently of their IQ-score, perinatal factors and parental characteristics. On average, the children with MHP also carry higher health

data, and cannot be shared publicly due to existing data protection laws in Denmark, and imposed by the Danish Data Protection Agency. The collected data is uploaded and linked to data from the Danish National registers and analysed through Statistics Denmark (https://www.dst.dk/en). Interested researchers can access the data used in this study free of cost through a study description and research agreement with the Copenhagen Child Cohort 2000 steering committee, by way of contact Pia Jeppesen at pia.jeppesen@regionh.dk or Allan Linneberg at allan.linneberg@regionh.dk.

**Funding:** PJ and RTW received funding from Commercial Company: Trygfonden, https://www.trygfonden.dk. AMS (The CCC2000 Study Group) received funding from Commercial Company: Lundbeckfonden, https://www.lundbeckfonden.com. AMS (The CCC2000 Study Group) received funding from Commercial Company: Trygfonden, https://www.trygfonden.dk. The funders have no role in study design, data collection and analysis, decision to publish, or preparation of the manuscript.

**Competing interests:** The authors have declared that no competing interests exist.

care costs over a five-year follow-up period. The higher health care costs are only attributed to 23% of these children, while the remaining children with MHP also show poorer school performance but receive no additional health care.

## Conclusions

The results demonstrate that children with MHP and a poor prognosis can be identified by the use of the brief standardised questionnaire SDQ combined with a screening algorithm.

## Introduction

Today more than 13% of children and adolescents worldwide are affected by mental disorders at any given time [1] with a significantly negative impact on their quality of life [2]. Several studies show that children's mental health problems lead to a higher risk of mental disorders later in life [3–9]. The median age of onset of anxiety, mood and behavioural disorders occur before age 14 years [10]. Despite the progress in the development and positive evaluation results of early treatment programs for mental health problems in childhood [11,12], such programs are not offered systematically and on a wide scale in Denmark or, to the best of our knowledge, in any other countries.

There is a growing awareness of the potentials of preventive interventions to improve children's mental health [11]. Specifically, on the potential of implementing them in 'stepped-care models' that offer interventions on increasing intensity levels ranging from general counseling to specialised mental health care, and with indicated prevention and early treatments as an intermediate step to children with common emotional and behavioral problems [13]. A systematic and wide-scale implementation of a stepped care model for service delivery needs to consider the complexities of childhood mental health problems. Children usually do not present with a recent-onset and well-defined single disorder. More commonly, the children have a long history of several problems, distress and impairments below or above diagnostic thresholds [13]. Thus, the 'matching' of the intervention to the children's fluctuating and complex problems is not straight forward. A systematic approach to identifying the children in need of interventions that are beyond prevention and counseling is therefore a necessary part of a successful stepped-care model.

For more than a decade it has been discussed how to identify mental health problems at an early stage among school children [14]. As outlined in a recent review there exists several studies evaluating different screening programs [15]. However, the quality of these evaluations is poor. The majority of the evaluations are based on cross-sectional study designs, and the few existing cohort studies include short follow-up periods and/or are dependent on self-reported outcomes [15]. Furthermore, the existing evaluations focus on diagnosis or access to mental health care. This could be problematic outcomes, as not all children in need of mental health care are likely to receive it [16–18]. Moreover, children with problems below the threshold for diagnosis may still require mental health care to avoid major difficulties later in life [9,19].

This study evaluates the potential of a new screening algorithm to identify children with sub-threshold and undiagnosed mental disorders, in this study referred to as mental health problems (MHP), in a general population. A unique combination of historical birth cohort data and high-quality data from national registers is used to mimic the performance of screening by analysing the later school performance and health care utilisation of the screened children. As neither parents, teachers nor health care professionals receive any feedback on the

screening this allows for analyses of how the children who screen positive perform in school up to five years later and how they utilise the health care system in a naturalistic follow-up. The rich data allows analyses that controls for potential confounders, i.e. the children's IQ, perinatal factors and their parents' health care utilisation and socio-economic characteristics. The methods section provides the precise definitions of the potential confounders.

The screening is based on parent-reported answers to the Strength and Difficulties Questionnaire (SDQ) and an algorithm, that combines scores of emotional and behavioural problems with functional impairments and uses cut-off based on population studies. The children's later school performance is investigated as a primary indicator of the long-term outcome of the screening for MHP in preadolescence. As previous studies find an association between children's MHP and their school performance [20–22], and school performance and schooling are known to be comprehensive predictors for the rest of a child's life with respect to both wealth, health, and happiness [23–25], this indicator is well-suited for the evaluation of the screening. The children's health care service utilisation during the five-year follow-up period is analysed for two purposes. First, to investigate whether and when the children that screen positive for MHP get access to specialised mental health care. Thereby the potential for early treatment can be assessed. Second, to investigate whether a different health care utilisation pattern is observed for the children that screen positive in the health care system.

## Materials and methods

### Ethics approval and consent to participate

All procedures performed in studies involving human participants were in accordance with the ethical standards of the institutional and/ or national research committee and with the 1964 Helsinki declaration and its later amendments or comparable ethical standards. The study was approved by The Danish Data Protection Agency (J.nr. 2010-41-4438) and by the Capital Region of Denmark (J.nr. 2007-58-0015). The National Committee on Health Research Ethics was consulted (J.nr. H-C-FSP-2010) in accordance with national guidelines. Participation was voluntary, and data was kept confidential. According to Danish legislation, individual informed consent or additional permission from the National Committee on Health Research Ethics is not required for registry-based studies. The parents gave oral informed consent to the study participation. Separate written consent was not required by the ethical committee. Moreover, the written information given to the participants stated that the participation in the study is voluntary, that by answering the questionnaires consent is given and consent can be withdrawn at any time.

### Data

**The Copenhagen Child Cohort.**   The Copenhagen Child Cohort (CCC2000) is a Danish birth cohort that includes all children born in 16 municipalities in the former Copenhagen County in the year 2000. In total, the cohort comprises 6,090 children, and constitutes 9% of all children born in Denmark that year. The cohort is representative of the general Danish population born that year in regards to both gender and a number of key perinatal factors such as age of parents and perinatal illness. Skovgaard et al. [26]provide further information on the cohort at birth. The main purpose of CCC2000 is to study the presentation, developmental pathways and risk mechanisms of psychopathology longitudinally from birth. As part of the follow up studies of the cohort, the parents of the children were invited to complete the "Strengths and Difficulties Questionnaire" (SDQ) at different points in time, i.e. when the children are aged 5–7 and 11–12 years. The present study uses SDQ responses from when the

children were aged 11–12 years. In total, parents of 2,126 children (35%) completed SDQ at the 11–12 years follow-up.

**The screening algorithm.** We use SDQ to define whether or not a child screens positive for MHP. SDQ is widely used and validated as a tool to identify and assess children with MHP in both clinical samples and in general population-based samples [27–32]. SDQ contains 25 items, which cover five subscales relating to the children's emotional problems, peer problems, behavioural problems, hyperactivity and pro-social behaviour. Responses to the subscales on emotional problems, peer problems, behavioural problems, and hyperactivity can be used to calculate a total difficulties score. Each subscale score ranges from 0–10, implying that the total difficulties score ranges from 0 to 40 [33].

We use an extended version of SDQ, which also includes an impact assessment to evaluate how much the identified mental difficulties interfere with the child's everyday life. A functional impairment score is calculated from five items on whether the difficulties upsets or distresses the child and how much the difficulties interfere with home life, friendships, classroom learning, and leisure activities. Each item is scored on a scale from 0 to 2. To score 1 or 2, the interference from the difficulties in that domain must be assessed to either "quite a lot" or "a great deal" [28,34].

The present study defines MHP as having a functional impairment score of at least 1 in combination with at least one of the following scores: a total difficulties score of at least 14 and/or an emotional problem score of at least 5 and/or a behavioural problem score of at least 3. This definition is based on age-matched populations in Denmark, Germany and the UK [35–37], such that scoring above the cut-off implies that the children's mental health state is ranked among the 10% of the poorest in these populations. This definition of MHP is also used in an on-going Danish randomized, clinical trial of an early treatment intervention for children with MHP (Trial ID: NCT03535805).

Based on the above criterion, we define children as either having MHP (screening positive) or not (screening negative) at time of reporting, i.e. at age 11–12 years. Thus, we do not distinguish between different levels of MHP in this study.

**School performance.** In Denmark, it is compulsory to attend school for nine years, from the age of 7 years to 15 years. The majority of Danish children attend public schools, which are fully funded through taxes. Private schools are also heavily subsidised. We retrieve information on each child's attendance and grades for each final exam from their compulsory schooling from the Danish Education Registers [38]. This information includes test scores up until summer 2017, where the children were between the age of 16.5 years and 17.5 years and thus, expected to have finished compulsory schooling.

At the end of compulsory schooling (9th grade), there are eight mandatory exams for all pupils distributed across four subjects: Danish, Mathematics, English and Physics/Chemistry. Following the Danish Ministry of Education that uses at least four exams to calculate the national Grade Point Average (GPA), we classify children who have not attended at least four of the eight mandatory exams as not having completed the 9th grade exams. Each exam results in a grade based on the Danish 7-point grading scale. The 7-point scale consists of seven grades: −03, 00, 02, 4, 7, 10, 12. These are equivalent to the following ECTS grades: F, Fx, E, D, C, B, A [39]. We calculate the children's GPA from the 9th grade mandatory exams (results from at least four exams) as a school performance outcome.

**Health care utilisation.** In Denmark the health care system is predominantly publicly financed by taxes, and information on individuals' health care utilisation and associated unit costs are listed in administrative registers. This study defines health care utilisation as the costs incurred for both somatic and mental health care and in both the primary and secondary health care sector. As there is no available information on privately funded health care, we

only include the costs of publicly funded care, with the exception of out-of-pocket spending on prescription medicine.

We analyse health care costs for the years 2012 to 2016 (equivalent to the five years post "screening"). We analyse the health care costs both at an aggregated level as total health care costs and in subsets based on sectors and specific types of mental health care, as outlined in section 2.4.1–2.4.3. We adjust all costs to 2016-prices using price- and wage-regulations of consumption prices from the Danish Regions and Statistics Denmark [40,41]. We convert the costs from Danish kroner (DKK) to Euros (€) using the currency rate DKK 100 = € 13.42.

We retrieve the costs of health care in the primary health care sector from the Danish National Health Service Register [42]. The costs are based on providers' activity-based remuneration. We differentiate between specialised and non-specialised mental health treatment in primary care. Specialised mental health treatment includes services from psychiatrists and psychologists, whereas non-specialised mental health treatment includes services provided by all remaining primary health care providers, e.g. general practitioners.

We retrieve costs of prescription medication from the Danish National Prescription Registry [43]. This database contains information on all purchase of prescription medications. We use the retail price at the date of purchase to measure costs. Depending on the annual consumption, between 60% and 100% of the cost of prescription medication is subsidised. Hence, we analyse the total costs of prescription medication for each child, implying that we include both the public funding and out-of-pocket spending. As there is no available information on purchase of over-the-counter medication, we do not include costs of this type of medication.

Costs of hospital care are retrieved from the National Patient Register [44]. We differentiate between somatic care and mental health care in hospitals, as well as inpatient care and outpatient care. We compute costs of somatic care using the Danish System of Diagnostic Related Groups (DRG-tariffs). DRG-tariffs are unavailable for mental health care in Denmark. Instead, we use the National Board of Health's per diem charge and ambulatory charge for mental health care to assess the costs of mental health care in hospitals.

**Covariates.** School performance and health care utilisation may be affected by factors that may also be associated with MHP. In an effort to isolate the effect of MHP status on subsequent outcomes, the analyses include several covariates derived from Statistics Denmark's national registers. For the individual child, this includes dummies for the following: the child's gender, whether the child is the first-born in the family, whether the child was born with low-birth-weight (<2,400 gram) (LBW) and/or was small-for-gestational-age (SGA), whether the child is second-generation immigrant (there are non-first generation immigrants in the cohort), and whether the child lived with both biological parents the 1st of January 2012.

For both biological parents of the child, we include dummies for the following: whether the parent received mental health care in a hospital during the years 1995 and 2011, whether the highest education in the beginning of 2012 was primary school (up to 9 years education), whether the parent was unemployed in 2011 (defined as unemployed for at least six months), and whether the parent was among the 25% with the highest total health care costs in the CCC2000 population (split by gender) from 2000 to 2011.

Children's IQ is potentially a confounding variable when estimating the association between school performance and MHP. The Block Design (BD) test is a subtest of Wechsler Intelligence Scale for Children (WISC-III). The BD test is known to be highly correlated with IQ on the full Wechsler test of intelligence [45]. We therefore proxy the children's IQ using results from the BD test, which was done face-to-face for those children in the cohort who accepted the invitation. The tests were done at approximately the same time as SDQ responses were completed by the parent.

## Research design and data analysis

**Study population.**   The focus of this study is to evaluate the potential of the screening process, we therefore exclude children who had already received specialised mental health care (SMHC) before the date of the parent completing SDQ. SMHC is defined as any contact with psychologists or psychiatrists in the primary health care sector or any contact to a hospital mental health care department. In total, 111 children (5.2%) had received SMHC at some point prior to the parents completing SDQ. We exclude these children from the study population, resulting in a study population of 2,015 children.

**Statistical analyses.**   In an effort to isolate the effects of MHP on school performance and health care utilisation, we control for the covariates listed in Section 2.5. However, for most of these covariates, it remains unclear whether they confound an association between MHP, school performance and health care costs, or if the single covariate represents an intermediary step on a causal pathway. All analyses are therefore done using three different models: 1) a crude model, 2) a model that includes covariates for the child's characteristics, and 3) a model that includes covariates for both the child's and the parents' characteristics.

For analyses of school performance, we also include the BD test score (IQ-proxy) in separate models, yielding two additional models: 4) a model including the IQ-proxy and the additional covariates for the child's characteristics and 5) a "full model" including the child's IQ-proxy and additional covariates for both the child's and the parents' characteristics. A total of 1,537 children (76.3% of the study population) completed a face-to-face BD test. Hence, the models that include the IQ-proxy are based on this smaller study population.

We also analyse the association between MHP and the outcomes based on whether the children with MHP received SMHC during the five-year follow-up period. For the group of children with MHP that received SMHC, we analyse whether there is a "window of opportunity" for early treatment of the children.

The comprehensive number of covariates in some of the models increases the risk of "over controlling" and thereby, possibly underestimating the associations. We therefore expect the true estimate to be somewhere in the range of the estimates from the different models. By presenting results from all models, we aim to explore the robustness of any significant associations found.

We measure the children's school performance using two different outcome measures: 1) completion of the 9th grade exams and 2) the GPA from the 9th grade exams. We employ a logit regression to estimate the association of MHP with the completion of the exams, whereas we use an Ordinary Least Squares (OLS) regression to estimate the association between MHP and the GPA, which is measured on a continuous scale.

We measure the children's health care utilisation in different parts of the health care sector. These data are heavily right-skewed, and some outcomes have a high proportion of zeros (corresponding to no consumption). We therefore employ both Generalized Linear Models (GLM) and two-part models combining a logit model with a GLM to estimate the association between MHP and the different types of health care utilisation.

We specify the GLM models based on the performance of GLMs with normal, gamma, Gaussian, and inverse Gaussian distributions combined with linear, log, power (0.5), and power (-1) links. Following Deb et al. [46], we choose the link function and distribution using Akaike Information Criteria, link-test, and Park's modified test. For the analysis of the children's total health care costs and primary health care sector care costs (without psychologist/psychiatrist), a GLM with log-link and gamma distribution performs best. For the rest of the health care costs outcomes, two-part models combining a logit and GLM with log-link and gamma distribution performs best.

## Results

### Descriptive statistics

Table 1 describes the study population split by whether or not the child screened positive for MHP at age 11–12 years. We categorise 9.6% of the full study population as having MHP. Several covariates are statistically significantly associated with having MHP. This includes being the first-born child in the family (including only child), not living with both biological parents, the mother having had contact to mental health hospital services, the mother's highest educational attainment being primary school, the mother being unemployed, and either parent being among the 25% of parents with the highest health care costs in decade leading up to the screening. In the study population for which an IQ-proxy test score is available, the percentage of children with MHP is slightly lower than in the full study population.

Table 2 shows descriptive statistics for the outcome variables. A larger share of children with MHP do not manage to complete the 9[th] grade exams, and amongst those who do complete the exams, a lower GPA is attained. Furthermore, the average total health care costs from 2012 to 2016 is higher for children with MHP, and a higher share receive SMHC in the five-

**Table 1. Characteristics of the children and their parents in the study population, by their mental health problems status.**

|  | Children without mental health problems N = 1,821 (90.4%) | Children with mental health problems N = 194 (9.6%) |
|---|---|---|
| Girl | 51.8% | 50.5% |
| First-born in the family | 49.3% | 57.7% ** |
| Born with low-birth-weight or small-for-gestational-age | 5.9% | 8.3% |
| Second-generation immigrant[a] | 10.5% | 10.8% |
| Child did not live with both biological parents 1. Jan 2012 | 23.1% | 38.7% *** |
| Mother had contact to the mental health hospital services between 1995 and 2011 | 7.6% | 13.9% *** |
| Father had contact to the mental health hospital services between 1995 and 2011 | 5.4% | 7.7% |
| Mother's highest education primary school (up to 9 years) | 11.8% | 17.0% ** |
| Father's highest education primary school (up to 9 years) | 16.2% | 19.6% |
| Mother was unemployed in 2011 | 5.3% | 11.3% *** |
| Father was unemployed in 2011 | 4.5% | 7.2% * |
| Mother among the 25% of the CCC2000 mothers with highest health care costs between 2000 and 2011 | 20.2% | 30.4% *** |
| Father among the 25% of the CCC2000 fathers with highest health care costs between 2000 and 2011 in the study population | 23.0% | 29.9% ** |
| **N (study population with IQ-proxy)** | **1,395 (90.8%)** | **142 (9.2%)** |
| IQ proxy (Mean Block Design score (SE)) | 9.78 (0.10) | 9.18 (0.34) * |

Note: The table describes the characteristics of children and their parents in the study population by their children's mental health problem status. Differences in the characteristics across the children's mental problem status is tested using a Chi-square test and two-sided t-test for Block Design score.

[a]There are no first generation immigrants in the cohort.

*p-level <0.1

**p-level <0.05

***p-level <0.01.

**Table 2. School performance and health care utilisation of the study population based on their mental health problems status.**

| | N | Children without mental health problems N = 1,821 (90.4%) | Children with mental health problems N = 194 (9.6%) |
|---|---|---|---|
| Have not completed the 9th grade exams | 2,015 | 3.1% | 9.3% |
| Mean grade point average (GPA) 9th grade exams | 1,941 | 7.72 | 6.45 |
| Mean total health care cost (€) 2012–2016 | 2,015 | 4,785 | 10,829 |
| Received SMHC 2012–2016 | 2,015 | 5.6% | 22.7% |

Note: The table summarises the school performance and health care utilisation of the study population based on their mental health status. Having completed the 9th grade exams is defined by having attended at least four of the eight mandatory exams. Total health care costs include all publicly funded care in both the primary and secondary health care sector. Specialised mental health care (SMHC) is defined as a contact with a publicly funded psychiatrist or psychologist.

year period. We present statistical tests results for associations between these outcomes and MHP in the following sections.

## School performance

The association between MHP and school performance is estimated using two different school outcomes. The first school outcome measures completion of the 9th grade exams. Table 3

**Table 3. The association between mental health problems at age 11–12 years and not completing the 9th grade exams.**

| | N | Mental health problems at age 11–12 years | |
|---|---|---|---|
| | | OR | SE |
| Crude model | 2,015 | 3.22 *** | 0.91 |
| Model including child's characteristics[a] | 2,015 | 3.14 *** | 0.91 |
| Model including child's characteristics[a] and IQ-proxy | 1,537 | 1.71 | 0.68 |
| Model including child's and parent's characteristics[a,b] | 2,015 | 3.00 *** | 0.87 |
| Model including child's, parent's characteristics[a,b], and IQ-proxy | 1,537 | 1.57 | 0.65 |

Note: The table contains estimated Odds Ratios (OR) of having a mental health problem at age 11–12 years and not completing the 9th grade exams. Logistic regression models are used to estimate OR with standard error (SE). As not all children have an IQ-test score the model including this population has fewer observations. We define not having completed the 9th grade exams as not attending at least four of the eight mandatory exams. Estimates for all covariates are available in Table A in S1 Table.

[a] Gender, first-born child in the family, born small-for-gestational-age or with low-birth-weight, children are second generation immigrants, child did not live with both biological parents 1st January 2012.

[b] Mother and/or father had contact to mental health hospital services between 1995 and 2011, mother's and/or father's highest education on 1st January 2012 was primary school (up to 9 years schooling), mother and/or father was unemployed in 2011, mother and/or father was among the parents of the full CCC2000 cohort with the 25% highest health care cost between 2000 and 2011.

*p-level <0.1

**p-level <0.05

***p-level <0.01.

shows the results. The three models without the IQ-proxy suggest that children with MHP have statistically significant higher odds of not completing the 9th grade exams. This difference is, however, statistically insignificant when controlling for the children's IQ.

The second school outcome is the GPA for the children, who have completed the 9th grade exams. Table 4 shows a statistically significant negative association between MHP and the GPA. Across models, the marginal effect ranges from -0.98 grade point to -1.27 grade point.

## Health care utilisation

Table 5 presents the results of the analyses of the association between MHP and the health care utilisation in the subsequent five years after the SDQ-screening (2012 to 2016). The association between MHP and total health care utilisation is statistically significant and ranges from €5,858 to €7,336 across models. We split the total health care utilisation into seven subcategories. The analyses of the subcategories provide information on which type of health care service drives the differences in the total health care costs. The results show that the higher costs for children with MHP are primarily driven by higher inpatient and outpatient mental hospital care. However, costs related to prescription medicines and utilisation in the primary sector, both specialised mental care and other types of care, are also positively associated with MHP. There is no statistically significant association found between MHP and utilisation of somatic hospital care during the five-year period of analysis.

Table 5 also shows that the results are robust across all models. The only exception is the association between MHP and utilisation of mental inpatient hospital care, which is statistically insignificant when controlling for both children's and parents' characteristics. This

**Table 4. The association between mental health problems at age 11–12 years and Grade Point Average at 9th grade exams.**

| | N | Mental health problems at age 11–12 years | |
| --- | --- | --- | --- |
| | | ME | SE |
| Crude model | 1,941 | -1.27 *** | 0.19 |
| Model including child's characteristicsa | 1,941 | -1.13 *** | 0.18 |
| Model including child's characteristicsa and IQ-proxy | 1,486 | -1.03 *** | 0.19 |
| Model including child's and parent's characteristicsa,b | 1,941 | -1.04 *** | 0.18 |
| Model including child's, parent's characteristicsa,b, and IQ-proxy | 1,486 | -0.98 *** | 0.19 |

Note: The table contains estimated coefficients (in grade points) for the association between mental health problems at age 11–12 years and Grade Point Average (GPA) at 9th grade exams. The study population includes children who have done at least four of the 9th grade exams. As not all children have an IQ-test score the model including this population has fewer observations. Ordinary Least Square regression models are used to estimate Marginal effect (ME) with robust standard error (SE) of mental health problems on GPA. Estimations for all covariates are available in Table B in S1 Table.

a Gender, first-born child in the family, born small-for-gestational-age or with low-birth-weight, children are second generation immigrants, child did not live with both biological parents 1st January 2012.

b Mother and/or father had contact to mental health hospital services between 1995 and 2011, mother's and/or father's highest education on 1st January 2012 was primary school (up to 9 years schooling), mother and/or father was unemployed in 2011, mother and/or father was among the parents of the full CCC2000 cohort with the 25% highest health care cost between 2000 and 2011.

*p-level <0.1

**p-level <0.05

***p-level <0.01.

**Table 5. The associations between mental health problems at age 11–12 years and health care costs in the subsequent five years.**

| | Crude models | | Models including child's characteristics[c] | | Models including child's and parent's characteristics[c,d] | |
|---|---|---|---|---|---|---|
| | Mental health problems at age 11–12 years | | | | | |
| | ME | SE | ME | SE | ME | SE |
| Total health care costs[a] | 6,044 *** | 2,254 | 7,336 *** | 2,301 | 5,858 *** | 1,713 |
| Primary sector care without psychologist/psychiatrist[a] | 180 ** | 73 | 176 ** | 72 | 149 ** | 72 |
| Psychologist/psychiatrist in primary care sector[a] | 202 * | 104 | 171 ** | 73 | 280* | 161 |
| Prescription medicine[b] | 161 ** | 78 | 140 * | 77 | 148 ** | 71 |
| Mental outpatient hospital care[b] | 717 *** | 208 | 772 *** | 223 | 696 *** | 222 |
| Mental inpatient hospital care[b] | 4,395 ** | 1,965 | 4,755 ** | 1,940 | 12,413[e] | 13,872 [e] |
| Somatic outpatient hospital care[b] | 332 | 262 | 324 | 250 | 182 | 219 |
| Somatic inpatient hospital care[b] | 55 | 609 | 686 | 660 | 904 | 690 |

Note: The table contains estimated coefficients (in €) of the association between mental health problems at age 11–12 years and health care costs in the five year-period. The number of observations N = 2,015. Estimations for all covariates in the analyses of total health care costs are available in Table C in S1 Table.

[a] Marginal effect (ME) and robust standard error (SE) estimated with a GLM model.

[b] Marginal effect (ME) and robust standard error (SE) estimated with a two-part model combining a logit and a GLM model.

[c] Gender, first-born child in the family, born small-for-gestational-age or with low-birth-weight, children are second generation immigrants, child did not live with both biological parents 1st January 2012.

[d] Mother and/or father had contact to mental health hospital services between 1995 and 2011, mother's and/or father's highest education on 1st January 2012 was primary school (up to 9 years schooling), mother and/or father was unemployed in 2011, mother and/or father was among the parents of the full CCC2000 cohort with the 25% highest health care cost between 2000 and 2011.

[e] Several variables were omitted in the GLM part of the two-part model due to lack of observations.

*p-level <0.1

**p-level <0.05

***p-level <0.01.

finding can, however, be explained by the relatively few children utilising mental inpatient hospital care compared to the large number of covariates in the model, which generates a large standard error.

**Health care utilisation and specialised mental health care treatment.** Table 5 shows that SMHC utilisation is the main driver of the higher health care utilisation of children with MHP. However, as reported in Table 2, only 22.7% of the children with MHP receive SMHC during the five-year follow-up period. To explore whether the children with MHP, who did not receive SMHC, receive more of other types of health care services, we split the analysis of the health care costs of children with MHP by whether or not they received SMHC. In these analyses, the reference group is still the "children with no MHP".

Table 6 shows the results of the crude models, i.e. without controlling for child and parental characteristics. We find that the higher total health care costs for the whole group of children with MHP is driven by the group of children with MHP who received SMHC. Besides generating high costs through their use of SMHC, this group of children also have higher costs due to utilisation of the primary health care sector and prescription medicine, and they tend to use somatic outpatient health care services more. The larger group of children with MHP who did not receive SMHC (77.3%) do not receive more health care services of any kind compared to

**Table 6. The associations between mental health problems at age 11–12 years (with/without specialised mental health care) and health care cost in the subsequent five years.**

| | Crude models | | | |
| --- | --- | --- | --- | --- |
| | Children with mental health problems who receive no specialised mental health care (n = 150) | | Children with mental health problems who receive specialised mental health care (n = 44) | |
| | ME | SE | ME | SE |
| Total health care costs[a] | -918 | 930 | 29,778 *** | 8,325 |
| Primary sector care without psychologist/psychiatrist[a] | 126 | 84 | 365 *** | 139 |
| Psychologist/psychiatrist in primary care sector[b] | NA | NA | 1,004 ** | 435 |
| Prescription medicine[b] | -18 | 62 | 775 *** | 244 |
| Mental outpatient hospital care[b] | NA | NA | 3,853 *** | 727 |
| Mental inpatient hospital care[b] | NA | NA | 22,043 *** | 7,917 |
| Somatic outpatient hospital care[b] | 157 | 292 | 929 * | 508 |
| Somatic inpatient hospital care[b] | -166 | 673 | 809 | 828 |

Note: The table contains estimated coefficients (in €) of the association between mental health problems at age 11–12 years and health care costs in the five year-period. We run separate analysis based on whether or not the children with MHP receive specialised mental health care. The reference group is children with no MHP. The number of observations N = 2,015. Some estimates are non-applicable (NA) based on the definition of the group. Specialised mental health care (SMHC) is defined as contact with publicly funded psychiatrist or psychologist. The findings are robust to including controls for both child and parental characteristics, see Table D in S1 Table.

[a] Marginal effect (ME) and robust standard error (SE) estimated with a GLM model.

[b] Marginal effect (ME) and robust standard error (SE) estimated with a two-part model combining a logit and a GLM model.

* p-level <0.1

** p-level <0.05

*** p-level <0.01.

children with no MHP. The findings are robust to including controls for both child and parental characteristics, see Table D in S1 Table.

**School performance and specialised mental health care treatment.** The results of the analyses of health care utilisation show that 77.3% of the children with MHP do not receive any SMHC and do not utilise the health care system more than children without MHP during the five-year follow-up period. To investigate whether this subgroup of children with MHP have an unmet need for care, we use the children's later school performance as an indicator of the consequences of their MHP at the age of 11–12 years. Thus, finding a negative association between having MHP without receiving SMHC and later school performance, compared to children with no MHP, will indicate that there is an unmet need for care.

We compare the school performance between children without MHP and the two groups of children with MHP, who are divided by whether or not they received SMHC during the five-year follow-up period. Tables 7 and 8 show the results. The group of children with MHP who did not receive SMHC does not have higher odds of not completing the 9th grade exams. The models without IQ-proxy shows a tendency towards a higher risk, but this tendency is statistically insignificant when controlling for IQ. However, the group of children with MHP who received SMHC have statistically significant higher odds of not having completed the 9th grade exams. The estimated odds ratio is almost halved after controlling for IQ. The statistical significance level is, however, independent of the chosen model.

**Table 7. The associations between mental health problems at age 11–12 years with/without specialised mental health care and not having completed the 9th grade exams.**

| | N | Children with mental health problems who receive no specialised mental health care (n = 150/109)[c] | | Children with mental health problems who receive specialised mental health care (n = 44/33)[c] | |
|---|---|---|---|---|---|
| | | OR | SE | OR | SE |
| Crude model | 2,015 | 2.01 * | 0.74 | 8.10 *** | 3.22 |
| Model including child's characteristics[a] | 2,015 | 1.90 * | 0.72 | 8.99 *** | 3.61 |
| Model including child's characteristics[a] and IQ-proxy | 1,537 | 1.04 | 0.57 | 4.91 *** | 2.66 |
| Model including child's and parent's characteristics[a,b] | 2,015 | 1.93 * | 0.75 | 7.67 *** | 3.10 |
| Model including child's, parent's characteristics[a,b], and IQ-proxy | 1,537 | 1.00 | 0.56 | 4.18 *** | 2.26 |

Note: The table contains estimated Odds Ratios (OR) of having a mental health problem at age 11–12 years and not completing the 9th grade exam. Logistic regression models are used to estimate OR with standard error (SE) for not having completed the 9th grade exams. Not having completed the 9th grade exams is defined by not having done at least four of the eight mandatory exams. Specialised mental health care (SMHC) is defined as contact with publicly funded psychiatrist or psychologist.

[a] Gender, first-born child in the family, born small-for-gestational-age or with low-birth-weight, children are second generation immigrants, child did not live with both biological parents 1st January 2012.

[b] Mother and/or father had contact to mental health hospital services between 1995 and 2011, mother's and/or father's highest education on 1st January 2012 was primary school (up to 9 years schooling), mother and/or father was unemployed in 2011, mother and/or father was among the parents of the full CCC2000 cohort with the 25% highest health care cost between 2000 and 2011.

[c] The sample size is lower in models which include IQ-proxy.

*p-level <0.1

**p-level <0.05

***p-level <0.01.

**Table 8. The associations between mental health problems at age 11–12 years with/without specialised mental health care and Grade Point Average at 9th grade exams.**

| | N | Mental health problems and no specialised mental health care (n = 141/105)[c] | | Mental health problems and specialised mental health care (n = 35/29)[c] | |
|---|---|---|---|---|---|
| | | ME | SE | ME | SE |
| Crude model | 1,941 | -1.29 *** | 0.21 | -1.17 *** | 0.45 |
| Model including child's characteristics[a] | 1,941 | -1.07 *** | 0.20 | -1.35 *** | 0.42 |
| Model including child's characteristics[a] and IQ-proxy | 1,486 | -0.95 *** | 0.21 | -1.31 *** | 0.42 |
| Model including child's and parent's characteristics[a,b] | 1,941 | -0.98 *** | 0.19 | -1.30 *** | 0.40 |
| Model including child's, parent's characteristics[a,b], and IQ-proxy | 1,486 | -0.90 *** | 0.20 | -1.26 *** | 0.41 |

Note: Ordinary Least Square regression models are used to estimate Marginal effect (ME) with robust standard error (SE) of mental health problems on Grade Point Average (GPA) at 9th grade exams. Specialised mental health care (SMHC) is defined as contact with publicly funded psychiatrist or psychologist.

[a] Gender, first-born child in the family, born small-for-gestational-age or with low-birth-weight, children are second generation immigrants, child did not live with both biological parents 1st January 2012.

[b] Mother and/or father had contact to mental health hospital services between 1995 and 2011, mother's and/or father's highest education on 1st January 2012 was primary school (up to 9 years schooling), mother and/or father was unemployed in 2011, mother and/or father was among the parents of the full CCC2000 cohort with the 25% highest health care cost between 2000 and 2011.

[c] The N is lower in models which include IQ-proxy.

*p-level <0.1

**p-level <0.05

***p-level <0.01.

In the analyses of GPA, shown in Table 8, we find a statistically significant negative association between MHP and GPA independently of whether or not the child has received SMHC. Amongst the children who have MHP but do not receive SMHC, the GPA score is between -0.90 and -1.29 grade points lower than non-MHP children, while it is between -1.17 and -1.35 grade points lower for the children with MHP who do receive SMHC.

**Window of opportunity and diagnoses.** To analyse the potential for earlier treatment of the children with MHP who receive specialised care within the five-year follow-up period, we analyse whether they receive the care with some delay compared to the timing of screening. The results include the 44 children with mental health problems and minimum one contact with SMHC. We find that the mean number of days between the parents' completion of SDQ and their children's first contact with SMHC was 945 days, and the median was 1,000 days. The 25% and 75% percentiles were 426 days and 1,311 days, respectively. Thus, the results show that both the average and median time span was more than two and a half years from when the parent responded to SDQ to the first contact with any SMHC.

Diagnoses are available for the children in the study population that received SMHC at a hospital. In total 16.8% of the children with MHP received a psychiatric diagnosis during the five-year follow-up period compared to 3.6% of the children we classify as not having MHP at age 11–12 years.

Of the 44 children with MHP receiving SMHC during the five-year follow-up period, 33 children (75%) received SMHC at a hospital. Thus, diagnoses are available for these children. They receive a broad variety of diagnoses: Hyperactivity disorders, adjustment reactions, pervasive developmental disorders, depressive disorders, psychotic disorders, and anxiety disorders are among the most frequent diagnoses. To comply with the data protection regulation, we refrain from reporting all specific diagnosis for this relatively small sample of individuals.

## Discussion

Using parents' responses to SDQ and the screening algorithm, we categorise 9.6% of the 11-12-year-old children in our study population as having MHP. We find that the children with MHP independent of potential confounders perform statistically significantly worse at the 9[th] grade exams, which are conducted about five years after the screening. This finding is in line with previous studies that have found an association between SDQ scores and school outcomes [21,22]. The results indicate that the screening algorithm succeed in identifying children that may benefit from an early intervention.

Children with MHP also exhibit a statistically significant higher utilisation of health care during the five-year follow-up period. Sub-group analyses show that the higher health care utilisation is entirely driven by the 22.7% of the children with MHP who receive SMHC during the follow-up period. The children we identified with MHP, who receive SMHC during the follow-up period, have their first contact with specialised care on average 2.5 years after the parent's completion of SDQ. This indicates that there for the majority of the group is a 'window-of-opportunity' for an early intervention.

The 77.3% of children with MHP who do not receive any SMHC during the follow-up period also obtain a statistically significantly worse GPA at their 9[th] grade exams than children with no MHP. However, these children do not receive any other additional health care services, which indicates that they have unmet need for care throughout the follow-up period.

We find that the children with MHP that receive SMHC have worse school outcome than the children with MHP that do not receive SMHC. This finding should, however, not necessarily be interpreted as the SMHC having a negative effect on the children's long-term outcome. It is more likely that the difference is due to confounding by indication, meaning that those

who are affected the most from the MHP also are more likely to get access to SMHC doing the follow-up.

The results show that the screening algorithm combined with parental-responses to SDQ enables us to identify of a group of children with MHP and a poor prognosis based on their later school performances. In Denmark and all other countries, to the best of our knowledge, evidence-based early treatment programs are not being offered systematically. The results of our study indicate that when no systematic screening is in place, the majority of the children with MHP receive no care despite of a poorer prognosis. Furthermore, those who get access to SMHC get it with a delay, potentially due to the MHP having progressed in severity thus making them eligible for SMCH. In a future implementation of a stepped-care model a screening procedure, like the one presented, could play an important first step in the 'matching' of intervention and severity of MHP.

## Strengths and limitations

A key strength of this study is the unique combination of cohort data including the face-to-face IQ-proxy test combined with the full register data on the children's school performance and health care utilisation, as well as information on the broad range of socio-economic and health-related covariates related to both the children and their parents. Despite the small population groups in some analyses, the results remain robust in all analyses independent of the inclusion of potential confounders, which further strengthens our findings.

There are, however, also some limitations. Due to data limitations, we were unable to include costs of possible visits and treatments at privately funded psychiatrists and psychologist. As publicly funded counselling or preventive measures vary across municipalities and are not systematically reported, we were also unable to include this information. However, the municipalities do not provide any systematic evidence-based practices. Their services can be considered as a first step in a stepped-care model and do not have the characteristics of early treatment. The missing information will therefore not change any of our conclusions. As we miss the information on private and municipality costs, the cost estimates should, however, be interpreted from a public health care sector perspective.

The study population is part of the CCC2000, which is representative of the Danish population on several key characteristics. However, as our study population is limited to children whose parents' respond to SDQ, our sample is not representative of the general population. On average, the study's parents have a higher educational attainment, are less unemployed, and have fewer contacts to mental health care services than the general population (see Table E in S1 Table). We do, however, not see any reason for our findings to be less relevant for children with parents from lower socioeconomic groups. As these parents are likely to have fewer resources to cope with the challenges of MHP, the negative impact of MHP on children's school performance could possibly be even worse for this group. Our sample size does not allow for comprehensive subgroup analysis of possible effect modifications between socioeconomic factors and MHP.

## Screening tool and timing

This study evaluates the potential for identifying children, who could benefit from an early intervention, using the SDQ and a new screening algorithm. This approach has several appealing properties. First SDQ is relatively brief and can easily be distributed and quickly answered by parents. Secondly, the scoring of SDQ allows an algorithm that combines symptoms scores with the impairment score such that we can identify children both with a certain load of symptoms and who are affected in their daily lives. Third, SDQ can be used to assess a broad range

of MHP. Thus, we identify a wide variety in types of diagnosis among the children that screened positive and received a mental disorder diagnosis doing the follow-up. It is, however, possible that other instruments and/or more sophisticated screening algorithms could be as good or better at identifying children in the need of early intervention for MHP.

This study bases its results on a screening of children at the age of 11–12 years. SDQ data are also available for the CCC2000 from when the children were aged 5–7 years (pre-school age). The choice of using the data from when the children were aged 11–12 years is based on the availability of the IQ-proxy test score. In an additional analysis, we estimate the association between MHP at aged 5–7 years (using the same SDQ cut-off, we identify 5.6% of the children with MHP) and later school performance without controlling for IQ and health care utilisation. Table F and G in S1 Table, report the results for children with MHP at age 5–7 years and their later outcomes.

We do not find any association between MHP identified at pre-school age and the children's school performance in 9th grade. For health care utilisation, we find results similar to the ones we report for the children identified at age 11–12 years. More specifically, MHP at the pre-school age is associated with higher health care costs in the subsequent 11 years. The estimates are, however, statistically significantly lower than the ones we find when screening the older age group. Although far from conclusive, these findings suggest that the age of screening is an important factor, and that our definition of MHP might not be as predictive among a pre-school population. Future studies should investigate the age factor further, and studies focusing on early treatment should consider this factor in the identification of children with MHP.

## Conclusions

This study provides evidence that supports the potential for systematically early identification of MHP among children. Future studies should conduct a more comprehensive psychopathological assessment of the children identified with the presented SDQ algorithm, to give a further understanding of the full potential of systematically early identification of MHP and subsequently early treatment.

## Supporting information

**S1 Table. Contains seven additional tables with results (Table A-G).**
(DOCX)

## Acknowledgments

We thank Anja Munkholm, Anne Mette Skovgaard, Charlotte Ulrikka Rask, Else Marie Olsen, Hanne Elberling, Lars Clemmensen, and Martin K. Rimvall from the co-author group "The CCC2000 Study Group".

## Author Contributions

**Conceptualization:** Rasmus Trap Wolf, Pia Jeppesen, Dorte Gyrd-Hansen, Anne Sophie Oxholm.

**Data curation:** Rasmus Trap Wolf, Anne Sophie Oxholm.

**Formal analysis:** Rasmus Trap Wolf.

**Funding acquisition:** Rasmus Trap Wolf, Pia Jeppesen.

**Investigation:** Rasmus Trap Wolf, Pia Jeppesen.

**Methodology:** Rasmus Trap Wolf, Pia Jeppesen.

**Project administration:** Rasmus Trap Wolf.

**Supervision:** Pia Jeppesen, Dorte Gyrd-Hansen, Anne Sophie Oxholm.

**Validation:** Rasmus Trap Wolf, Pia Jeppesen.

**Writing – original draft:** Rasmus Trap Wolf.

**Writing – review & editing:** Rasmus Trap Wolf, Pia Jeppesen, Dorte Gyrd-Hansen, Anne Sophie Oxholm.

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
