## [Decision Letter · Decision Letter 0]

10 Sep 2019

[EXSCINDED]

PONE-D-19-21674

Evaluation of a screening algorithm using the Strengths and Difficulties Questionnaire to identify children with mental health problems: A five-year register-based follow-up on school performance and healthcare use

PLOS ONE

Dear Rasmus Trap Wolf,

Thank you for submitting your manuscript to PLOS ONE. After careful consideration, we feel that it has merit but does not fully meet PLOS ONE’s publication criteria as it currently stands. Therefore, we invite you to submit a revised version of the manuscript that addresses the points raised during the review process.

We would appreciate receiving your revised manuscript by Oct 25 2019 11:59PM. To enhance the reproducibility of your results, we recommend that if applicable you deposit your laboratory protocols in protocols.io, where a protocol can be assigned its own identifier (DOI) such that it can be cited independently in the future. For instructions see: http://journals.plos.org/plosone/s/submission-guidelines#loc-laboratory-protocols

We look forward to receiving your revised manuscript.

Kind regards,

Eduardo Fonseca-Pedrero, PhD

Academic Editor

PLOS ONE

Journal Requirements:

Reviewers' comments:

Reviewer's Responses to Questions

**Comments to the Author**

1. Is the manuscript technically sound, and do the data support the conclusions?

Reviewer #1: Yes

Reviewer #2: Yes

2. Has the statistical analysis been performed appropriately and rigorously? 

Reviewer #1: Yes

Reviewer #2: Yes

3. Have the authors made all data underlying the findings in their manuscript fully available?

Reviewer #1: Yes

Reviewer #2: No

4. Is the manuscript presented in an intelligible fashion and written in standard English?

Reviewer #1: Yes

Reviewer #2: Yes

5. Review Comments to the Author

Reviewer #1: The paper has an important justification of the study, highlights the convenience of evaluate the potential of a new screening algorithm to identify children with sub-threshold and undiagnosed mental disorders, in a general population. A combination of historical birth cohort data and high-quality data from national registers is used.

Some interesting results and discussion are added to the international scientific literature.

It is a good paper. However, the manuscript has some weakness, which must be improved:

p.1-2. “Today more than 13% of children and adolescents worldwide are affected by mental disorders at any given time (1) with a significantly negative impact on their quality of life (2).” The date of the two previous cited papers are 2015 and 2002. Please add more recent papers. I recommend some recent publications: Self-reported mental health in children ages 6–12 years across eight European countries. Eur. Child Adolesc. Psychiatry 2018, 27, 785–795 (http://dx.doi.org/10.1007/s00787-017-1073-0) or International and Spanish Findings in Scientific Literature about Minors’ Mental Health: Predictive Factors Using the Strengths and Difficulties Questionnaire (2019)( DOI: 10.3390/ijerph16091603).

p.44-45. Please do the same with the 9 cited papers, used to explain the recent situation? Please look for more recent published papers.

p.61 “There exists several studies evaluating different screening programs (15)”. Please add more references than one.

p.77 “i.e. the children’s IQ”. Please define IQ.

p.134. Please clarify the definition of MHP . You do not make differences between borderline and abnormal? (see https://www.sdqinfo.com/) Please clarify it in the manuscript.

p447. Maybe table 9 could be deleted and insert all the information of this table in a paragraph, directly in the text.

p463. Could you think about insert other international results in the discussion, in order to compare you results with other international papers or find a possible explanation? Maybe it would improve the paper. There is no one reference in discussion and conclusions. Please consider it.

Reviewer #2: The work entitled “Evaluation of a screening algorithm using the Strengths and Difficulties Questionnaire to identify children with mental health problems: A five-year register-based follow-up on school performance and healthcare use" is of great interest. The research is very stimulating; it contains new scientific knowledge and provides comprehensive information for further development of this productive line of research. The data are presented in a clear and easy-to-understand fashion, and the paper is, principally, well-argued and clearly worthy of publication. However, I have some comments to make that should be addressed before I recommend this manuscript for publication

From my point of the view, the introduction would benefit from the inclusion of more literature about the SDQ use in mental health screening worldwide. Considering that the study is about the mental health problems and the parents’ report about mental health, some other studies about this would be advisable. For instance: Bach et al., 2019; Navarro et al., 2019; Ortuño-Sierra, Aritio-Solana, and Fonseca-Pedrero, 2018.

In the results section, it is not clear to me why the authors say that: “To investigate whether this subgroup of children with MHP have an unmet need for care, we use the children’s later school performance as an indicator of the consequences of their MHP at the age of 11-12 years”. In my opinion, this should be at least explained in the introduction. Is it there more literature confirming that a decrease in school performance can be used as an indicator of a increase in mental health problems

In the discussion section authors state that: “The 77.3% of children with MHP who do not receive any SMHC during the follow-up period also obtain a statistically significantly worse GPA at their 9th grade exams than children with no MHP.” I do not clearly understand if the comparison should be between those who have MHP and those without MHP. It is not clear if the difference is because the lack of intervention or because of the fact that they have MHP. Authors explain this, but it seems obvious, so do not really see the point of including this.

6. PLOS authors have the option to publish the peer review history of their article (what does this mean?). If published, this will include your full peer review and any attached files.

Reviewer #1: No

Reviewer #2: No

---

## [Author Response · Author response to Decision Letter 0]

16 Sep 2019

Response to Reviewers

The following text is attached in a pdf-file as well.

Dear Editor,

We have made all the changes in the manuscript needed to meet PLOS ONE's style requirements.

We would like to thank the reviewers for their constructive and valuable comments and suggestions. These inputs have helped us significantly improve the manuscript by clarifying unclear points and referencing additional relevant studies. 

Below we address each reviewer’s comments separately. The changes made to the manuscript to address the reviewers’ comments is in red font (only in attached pdf).

Reviewer #1: 

Comment 1:

p.1-2. “Today more than 13% of children and adolescents worldwide are affected by mental disorders at any given time (1) with a significantly negative impact on their quality of life (2).” The date of the two previous cited papers are 2015 and 2002. Please add more recent papers. I recommend some recent publications: Self-reported mental health in children ages 6–12 years across eight European countries. Eur. Child Adolesc. Psychiatry 2018, 27, 785–795 (http://dx.doi.org/10.1007/s00787-017-1073-0) or International and Spanish Findings in Scientific Literature about Minors’ Mental Health: Predictive Factors Using the Strengths and Difficulties Questionnaire (2019)( DOI: 10.3390/ijerph16091603).

Thank you for the comment. Following the comment, we have re-reviewed the literature. We still find our two references to be highly relevant. Polanczyk et al. (2015) is the largest existing meta-study on prevalence of mental disorders and is well executed methodologically. Sawyer et al. (2002) is one of the few papers that studies children’s mental health disorders and health-related quality of life in a large representative sample. 

Comment 2:

p.44-45. Please do the same with the 9 cited papers, used to explain the recent situation? Please look for more recent published papers.

Following the reviewer’s suggestion, we have re-reviewed the literature, which has led us to include seven additional references in the introduction and methods section:

Line 52: 

9. Copeland WE, Wolke D, Shanahan L, Costello EJ. Adult Functional Outcomes of Common Childhood Psychiatric Problems: A Prospective, Longitudinal Study. JAMA Psychiatry. 2015;72: 892. doi:10.1001/jamapsychiatry.2015.0730

Line 75:

18. Roberts RE, Fisher PW, Blake Turner J, Tang M. Estimating the burden of psychiatric disorders in adolescence: the impact of subthreshold disorders. Soc Psychiatry Psychiatr Epidemiol. 2015;50: 397–406. doi:10.1007/s00127-014-0972-3

Line 77:

19. Balázs J, Miklósi M, Keresztény Á, Hoven CW, Carli V, Wasserman C, et al. Adolescent subthreshold-depression and anxiety: psychopathology, functional impairment and increased suicide risk: Adolescent subthreshold-depression and anxiety. J Child Psychol Psychiatry. 2013;54: 670–677. doi:10.1111/jcpp.12016

Line 94:

21. Kristoffersen JHG, Obel C, Smith N. Gender differences in behavioral problems and school outcomes. J Econ Behav Organ. 2015 Jul;115:75–93. 

22. Keilow M, Sievertsen HH, Niclasen J, Obel C. The Strengths and Difficulties Questionnaire and standardized academic tests: Reliability across respondent type and age. Goertz M, editor. PLOS ONE. 2019 Jul 25;14(7):e0220193. 

Line 133:

32. Ortuño-Sierra J, Aritio-Solana R, Fonseca-Pedrero E. Mental health difficulties in children and adolescents: The study of the SDQ in the Spanish National Health Survey 2011–2012. Psychiatry Res. 2018 Jan;259:236–42. 

Line 143:

34. Stringaris A, Goodman R. The Value of Measuring Impact Alongside Symptoms in Children and Adolescents: A Longitudinal Assessment in a Community Sample. J Abnorm Child Psychol. 2013;41: 1109–1120. doi:10.1007/s10802-013-9744-x

Comment 3:

p.61 “There exists several studies evaluating different screening programs (15)”. Please add more references than one.

Thank you for the comment. Our reference is a 2019 systematic review, it therefore covers existing studies evaluating different screening programs. To clarify, we added the following red text to the manuscript:

Line 70: As outlined in a recent review, there exists several studies evaluating different screening programs (15)

Comment 4:

p.77 “i.e. the children’s IQ”. Please define IQ.

Thank you for the suggestion. All variables are defined in the methods section. To clarify, we added the following red text to the manuscript:

Line 87-88: The methods section provides the precise definitions of the potential confounders.

Comment 5:

p.134. Please clarify the definition of MHP. You do not make differences between borderline and abnormal? (see https://www.sdqinfo.com/) Please clarify it in the manuscript.

Thank you for the suggestion. To clarify, we added the following red text to the manuscript:

Line 152-153: Thus, we do not distinguish between different levels of MHP in this study. 

Comment 6:

p447. Maybe table 9 could be deleted and insert all the information of this table in a paragraph, directly in the text.

Thank you for the suggestion. We have deleted table 9 and added the following red text to the manuscript:

Line 456-461: The results include the 44 children with mental health problems and minimum one contact with SMHC. We find that the mean number of days between the parents’ completion of SDQ and their children’s first contact with SMHC was 945 days, and the median was 1,000 days. The 25% and 75% percentiles were 426 days and 1,311 days, respectively. Thus, the results show that…

Comment 7:

p463. Could you think about insert other international results in the discussion, in order to compare you results with other international papers or find a possible explanation? Maybe it would improve the paper. There is no one reference in discussion and conclusions. Please consider it.

Thank you for the comment. To address this comment, we added the following red text to the manuscript:

Line 483-483: This finding is in line with previous studies that have found an association between SDQ scores and school outcomes [21,22]. The results indicate….

Reviewer #2: 

Comment 1:

From my point of the view, the introduction would benefit from the inclusion of more literature about the SDQ use in mental health screening worldwide. Considering that the study is about the mental health problems and the parents’ report about mental health, some other studies about this would be advisable. For instance: Bach et al., 2019; Navarro et al., 2019; Ortuño-Sierra, Aritio-Solana, and Fonseca-Pedrero, 2018.

Thank you for the comment. Following the comment, we have re-reviewed the literature including the suggested references. We have chosen to include those references that we believe have direct relevance to the paper. 

As mentioned in the response to reviewer #1 1e have added the following the following seven references to the introduction and methods section in the manuscript:

Line 52: 

9. Copeland WE, Wolke D, Shanahan L, Costello EJ. Adult Functional Outcomes of Common Childhood Psychiatric Problems: A Prospective, Longitudinal Study. JAMA Psychiatry. 2015;72: 892. doi:10.1001/jamapsychiatry.2015.0730

Line 75:

18. Roberts RE, Fisher PW, Blake Turner J, Tang M. Estimating the burden of psychiatric disorders in adolescence: the impact of subthreshold disorders. Soc Psychiatry Psychiatr Epidemiol. 2015;50: 397–406. doi:10.1007/s00127-014-0972-3

Line 77:

19. Balázs J, Miklósi M, Keresztény Á, Hoven CW, Carli V, Wasserman C, et al. Adolescent subthreshold-depression and anxiety: psychopathology, functional impairment and increased suicide risk: Adolescent subthreshold-depression and anxiety. J Child Psychol Psychiatry. 2013;54: 670–677. doi:10.1111/jcpp.12016

Line 94:

21. Kristoffersen JHG, Obel C, Smith N. Gender differences in behavioral problems and school outcomes. J Econ Behav Organ. 2015 Jul;115:75–93. 

22. Keilow M, Sievertsen HH, Niclasen J, Obel C. The Strengths and Difficulties Questionnaire and standardized academic tests: Reliability across respondent type and age. Goertz M, editor. PLOS ONE. 2019 Jul 25;14(7):e0220193. 

Line 133:

32. Ortuño-Sierra J, Aritio-Solana R, Fonseca-Pedrero E. Mental health difficulties in children and adolescents: The study of the SDQ in the Spanish National Health Survey 2011–2012. Psychiatry Res. 2018 Jan;259:236–42. 

Line 143:

34. Stringaris A, Goodman R. The Value of Measuring Impact Alongside Symptoms in Children and Adolescents: A Longitudinal Assessment in a Community Sample. J Abnorm Child Psychol. 2013;41: 1109–1120. doi:10.1007/s10802-013-9744-x

Comment 2:

In the results section, it is not clear to me why the authors say that: “To investigate whether this subgroup of children with MHP have an unmet need for care, we use the children’s later school performance as an indicator of the consequences of their MHP at the age of 11-12 years”. In my opinion, this should be at least explained in the introduction. Is it there more literature confirming that a decrease in school performance can be used as an indicator of a increase in mental health problems

Thank you for the comment. The aim of this section of the paper was to clarify that also the children who screen positive for MHP, but do not receive mental health care, may be affected by MHP. As studies find that school performance is an important proxy for future life prospects (21–23), we use school performance as a potential indicator of consequences of MHP. To clarify that the comparison is not between the two groups of children that screen positive for MHP, but between children that screen negative for MHP and children that screen positive for MHPs and do not receive mental health care, we added the following red text to the manuscript:

Line 406-408: Thus, finding a negative association between having MHP without receiving SMHC and later school performance, compared to children with no MHP, will indicate that there is an unmet need for care.

In addition, the following text in the background section (line 91-96) explains the choice of school performance as outcome:

The children’s later school performance is investigated as a primary indicator of the long-term outcome of the screening for MHP in preadolescence. As previous studies find an association between children’s MHP and their school performance [20–22], and school performance and schooling are known to be comprehensive predictors for the rest of a child’s life with respect to both wealth, health, and happiness [23–25], this indicator is well-suited for the evaluation of the screening.

Comment 3:

In the discussion section authors state that: “The 77.3% of children with MHP who do not receive any SMHC during the follow-up period also obtain a statistically significantly worse GPA at their 9th grade exams than children with no MHP.” I do not clearly understand if the comparison should be between those who have MHP and those without MHP. It is not clear if the difference is because the lack of intervention or because of the fact that they have MHP. Authors explain this, but it seems obvious, so do not really see the point of including this.

Thank you for the comment. The reason for describing the above is to highlight that, in comparison to children with no health problems, the majority of the children with mental health problems (the 77.3% who test positive for MHP but do not receive specialized care) have worse school outcome. This result suggests that it is not only the 22.7% that at some point receive specialized mental health care, who could potentially benefit from an early intervention. This result is described in line 487-488 in the manuscript.

---

## [Editor Report · Decision Letter 1]

19 Sep 2019

Evaluation of a screening algorithm using the Strengths and Difficulties Questionnaire to identify children with mental health problems: A five-year register-based follow-up on school performance and healthcare use

PONE-D-19-21674R1

Dear Dr. Rasmus Trap Wolf,

We are pleased to inform you that your manuscript has been judged scientifically suitable for publication and will be formally accepted for publication once it complies with all outstanding technical requirements.

With kind regards,

Eduardo Fonseca-Pedrero, PhD

Academic Editor

PLOS ONE
---

## [Editor Report · Acceptance letter]

26 Sep 2019

PONE-D-19-21674R1 

Evaluation of a screening algorithm using the Strengths and Difficulties Questionnaire to identify children with mental health problems: A five-year register-based follow-up on school performance and healthcare use.    

Dear Dr. Wolf:

I am pleased to inform you that your manuscript has been deemed suitable for publication in PLOS ONE. Congratulations! Your manuscript is now with our production department. 

With kind regards,

on behalf of

Dr. Eduardo Fonseca-Pedrero 

Academic Editor

PLOS ONE